

# Biomechanical analysis of iliosacral and transiliac–transsacral screw combinations for fixation of undisplaced Denis II vertical shear fractures in dysmorphic sacrum

Peishuai Zhao[1,*], Chengfei Peng[2,*], Honghu Lin[1], Ying Ji[1], Weiyi Pang[3] and Chaoyong Bei[1,4,5]

[1] Department of Orthopaedics, Guilin Medical University Affiliated Hospital, Guilin, Guangxi, China
[2] Department of Orthopaedics, Nanxishan Hospital of Guangxi Zhuang Autonomous Region, Guilin, Guangxi, China
[3] Guangxi Key Laboratory of Environmental Exposomics and Entire Lifecycle Heath, Guilin Medical University, Guilin, Guangxi, China
[4] Department of Biomedical Engineering, Guangxi Engineering Research Center of Digital Medicine and Clinical Transformation, Guilin, Guangxi, China
[5] School of Physical Education and Health, Guangxi Normal University, Guilin, Guangxi, China
[*] These authors contributed equally to this work.

## ABSTRACT

**Background**. Percutaneous iliosacral screws (ISS) and transiliac-transsacral screws (TTS) are effective for treating posterior pelvic ring instability. However, the biomechanical stability of undisplaced sacral dysmorphism fractures remains underexplored. This study evaluated various ISS and TTS combinations to provide a clinical reference for fixing such fractures.

**Methods**. A finite element model of a complete Denis type II dysmorphic sacral fracture (extending through the sacral foramen) was developed. The stability of the posterior pelvic ring was evaluated using seven fixation techniques: S1-ISS (Group 1), S2-ISS (Group 2), S1-ISS + S2-ISS (Group 3), S2-TTS (Group 4), anterior S1-ISS + S2-TTS (Group 5), middle S1-ISS + S2-TTS (Group 6), and posterior S1-ISS + S2-TTS (Group 7). In all models, the anterior pelvic ring was fixed with pubic ramus screws. The upper sacral surface was subjected to six loading modes to simulate physiological states: standing, forward flexion, left flexion, right flexion, left rotation, and right rotation. The following parameters were recorded and analyzed: vertical displacement and sagittal angular displacement of the upper sacral surface, relative displacement of five pairs of observation points on the anterior fracture line, and maximum Von Mises stress and deformation of the S1 and S2 screws.

**Results**. The finite element analysis revealed that the maximum stress in all internal fixation groups across the six loading modes was below the yield strength of titanium alloy, indicating no risk of implant failure. Screw deformation was highest in G1 and lowest in G5 for the S1 segment, and highest in G2 and lowest in G 5 for the S2 segment. G5 also exhibited the minimum vertical and angular displacements of the sacral upper surface in all motion states.

Corresponding author
Chaoyong Bei, beicy2023@163.com

**Conclusion**. This study demonstrates that S1-ISS combined with S2-TTS fixation provides excellent biomechanical stability for undisplaced vertical fractures of the sacral dysmorphism, particularly the combination of anterior S1-ISS and S2-TTS. This fixation method offers a promising clinical treatment option.

# INTRODUCTION

The sacrum is a critical component of the posterior pelvic ring stability, serving as the biomechanical link between the spine and the lower limbs (*Beckmann & Chinapuvvula, 2017*; *Roy-Camille et al., 1985*). Sacral fractures, often resulting from high-energy trauma such as motor vehicle accidents or falls from height, are frequently subjected to substantial vertical shear forces, which may lead to posterior pelvic ring instability (*Chung et al., 2020*; *Han et al., 2022*). According to research, sacral fractures comprise approximately 23.4%–30.4% of pelvic fractures (*Denis, Davis & Comfort, 1988*). Moreover, the incidence of traumatic sacral fractures has increased in recent years, largely due to industrial advancements (*Bydon et al., 2014*).

Presently, the mainstream surgical treatments for sacral fractures includes posterior plates (*Dienstknecht et al., 2011*), unilateral or bilateral triangular fixation (*Schildhauer et al., 2003*; *Tian, Chen & Jia, 2018*), iliosacral screws (ISS) (*Jäckle et al., 2023*), and transiliac-transsacral screws (TTS) (*Barger & Robinson, 2023*). Sacral fractures of Denis type II without displacement can maintain the relative position of the fracture ends due to the integrity and stability of the surrounding normal ligamentous structures. Satisfactory clinical outcomes are typically achieved after bed rest (*Matta & Tornetta 3rd, 1996*; *Tile, 1988*). However, ISS or TSS fixation techniques, which provide immediate stability through close contact with the bone tissue, can reduce micromotion at the fracture site. This approach allows for early mobilization, promotes early fracture healing, and reduces complications associated with prolonged bed rest (*Nork et al., 2001*). Studies have demonstrated that placing one TSS in each of the S1 and S2 vertebrae achieves optimal biomechanical stability (*Zhao et al., 2013*). However, studies have shown that some patients exhibit sacral dysmorphism, which can severely restrict the transverse screw trajectory due to the steep sacral wing slope (*Chen et al., 2024*; *Kim, Kim & Kim, 2022*). As a result, the S1 segment may not safely accommodate a TSS. Therefore, an oblique S1 ISS is often used to stabilize the posterior pelvic ring in cases without a viable TSS corridor (*Wendt et al., 2019*).

The placement of ISS is typically determined by combining sacral lateral views, pelvic inlet views, and outlet views to establish the direction and length of the screws (*Zhao et al., 2022*). In patients with sacral dysmorphism, the screw trajectory on the outlet view often tilts upward, while on the inlet view, it may be anterior, middle, or posterior. Studies have shown that placing the entry point of the screw slightly posterior and inferior to the S1 vertebral body projection allows for the safe placement of an oblique S1 ISS. The screw

trajectory in dysmorphic sacral is often "dumbbell-shaped", allowing for multiple positions and angles of oblique screws (*Cai et al., 2024*). Advances in imaging technology, such as intraoperative 3D CT navigation and robotic guidance, have provided reliable support for precise screw placement, reducing surgical risks (*Wan et al., 2023*). In-depth anatomical studies of the pelvis have also optimized the safe trajectory for screw placement, making the procedure more accurate, safe, and efficient (*Godolias et al., 2024*). However, there is currently a paucity of biomechanical studies on ISS for stabilizing sacral fractures with dysmorphism, particularly regarding the stability comparison of different orientations of S1-ISS combined with S2-TSS.

In this study, we intended to employ three-dimensional finite element technology to assess the biomechanical stability of various combinations of ISS and TTS used to treat sacral dysmorphism fractures (Denis type II), in order to give a reference for clinical application.

## METHODS

### Establishment of intact pelvic model

This study was approved by the ethics committee of Guilin Medical University Affiliated Hospital (Number: 2024QTLL-02) and used pelvic data from a healthy 31-year-old male (173 cm, 70 kg) with normal bone structure and no history of deformity, tumors, or surgery. The volunteer underwent a 64-slice CT scan (0.625 mm thickness). The data were processed in Mimics 21.0 to generate a complete pelvic model including bilateral hip bones and sacrum. Geomagic 17.0 was used for noise reduction, smoothing, and separation of cortical and cancellous bone, with iliac cortical thickness set at 1.5 mm and sacral cortical thickness at 0.45 mm. In Solidworks 2021, the cortical and cancellous bone were assembled, and sacroiliac and pubic symphysis cartilage were created. Ansys 21.0 was then used to assign material properties to bone and cartilage and to model ligaments based on prior studies (*Fu et al., 2014*). The final pelvic model contains 1,011,005 nodes and 613,587 elements (Fig. 1).

### Establishment of fracture model and internal fixation

A pelvic model with unilateral vertical instability injury, including a right-sided transforaminal sacral fracture and fractures of the upper and lower rami of the right pubis, was established through mesh line segmentation. Regardless of the direction of the oblique S1-ISS, the screw trajectory on the pelvic outlet view is from the outer inferior to the inner superior, although the screw trajectory on the inlet view might have an extensive angular range. We divided the probable needle insertion point area of the screw in the inlet view into three equal portions, which we defined as the anterior IS, the middle IS, and the posterior IS in different directions (Fig. 2). The ISS extend up to the midline of the S1 body, while the TTS connect one lateral external iliac cortex to the contralateral external iliac cortex. In seven experimental groups, posterior fixation was performed as follows: Group 1 (G1), S1-ISS; Group 2 (G2), S2-ISS; Group 3 (G3), S1-ISS + S2-ISS; Group 4 (G1), S2-TTS; Group 5 (G5), Anterior S1-ISS + S2-TTS; Group 6 (G6), Middle S1-ISS +

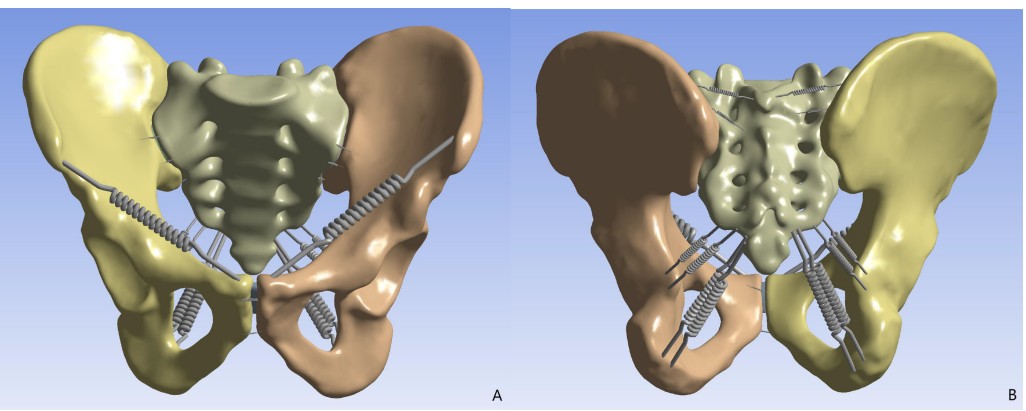

**Figure 1** **The finite element pelvic model.** (A) Anterior view; (B) posterior view.

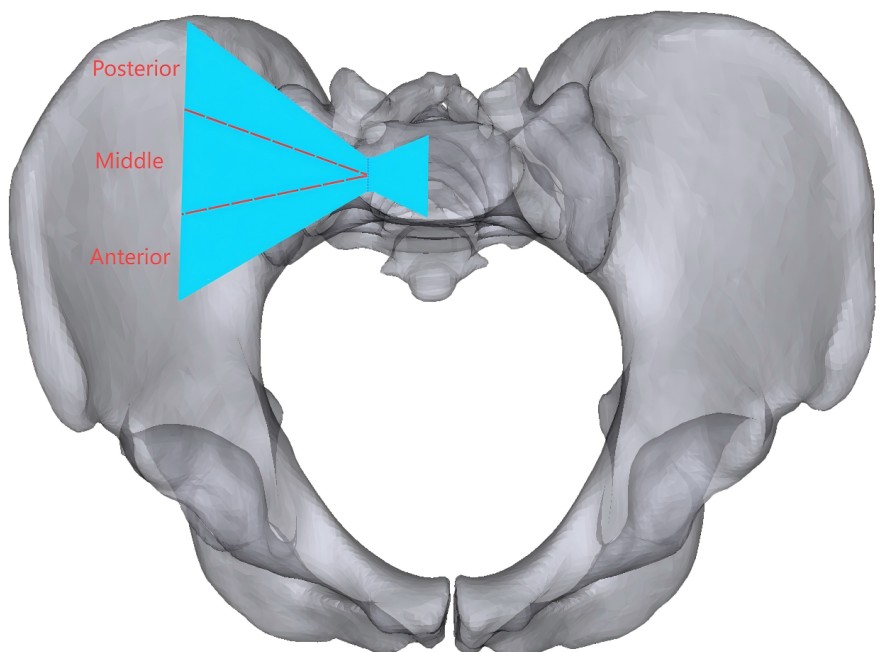

**Figure 2** **The oblique S1 iliosacral screw at the pelvic inlet view to the bone corridor are shown.** The blue area represents the bone corridor, which is divided into three equal portions based on the midpoint of the line across the narrowest part, and anterior, middle, and posterior ISS can be put in the anterior, middle, and posterior needle insertion point areas, respectively.

S2-TTS; Group 7 (G7), Posterior S1-ISS + S2-TTS (Fig. 3). The ISS and TTS both have a diameter of 6.5 mm.

## Loading of forces and setting of contact conditions

A vertical force of 500 N had been given to the upper surface of the sacral as a loading condition that imitated the upper body's weight, and the bilateral acetabular was fixed in six

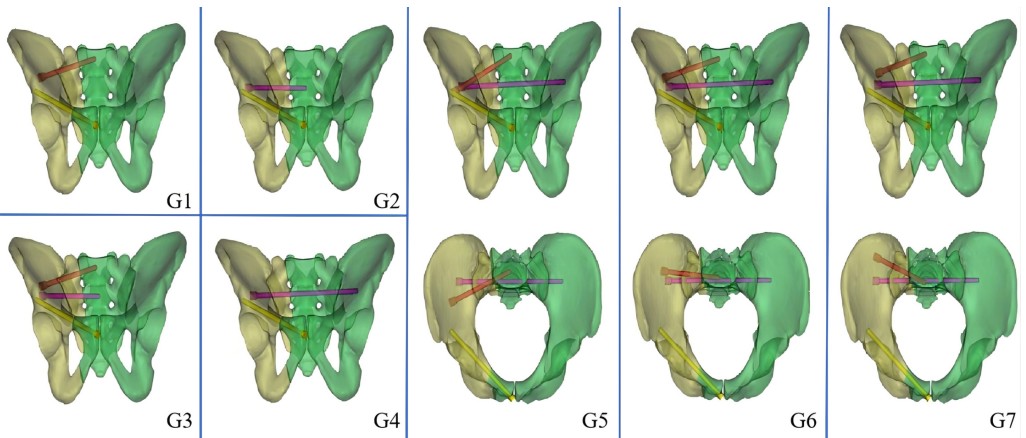

**Figure 3** **Model diagram of different internal fixation methods.** G1: S1-ISS; G2: S2-ISS; G3: S1-ISS+ S2-ISS; G4: S2-TTS; G5: Anterior S1-ISS + S2-TTS; G6: Middle S1-ISS screw + S2-TTS; G7: Posterior S1-ISS + S2- TTS.

directions. In addition, a vertical load of 500 N and a torque of 10 N m are applied to imitate the body's forward flexion, left flexion, right flexion, left rotation, and right rotation. In this finite element model, the contact conditions between cortical bone, cancellous bone, cartilage, and implants were set according to previous studies (*Gonçalves et al., 2023*).

### Assessment by finite element analysis

In this study, a number of parameters will be used to assess the fixation strength for different internal fixation methods. The parameters assessed included the maximum von Mises stress and screw deformation for S1 and S2 segmental screws, vertical displacement of the upper surface of the sacral, sagittal angular displacement of the upper surface of the sacral, and relative displacement of 5 pairs of observation points on the fracture line of the anterior surface of the sacral (Fig. 4).

## RESULTS

### Intact pelvic finite element model examination

In the process of finite element model validation in this study, we took research concepts from Miller and others (*Miller, Schultz & Andersson, 1987*; *Turbucz et al., 2023*; *Xu et al., 2020*). Specifically, five 294 N translational loads (upward, downward, forward, backward, and lateral) and three 42 N m rotational moments (flexion, extension, and rotation) were applied to the sacral upper surface. Displacements under these loads were measured to assess model reliability. The results closely matched those from previous studies, confirming the model's validity (Fig. 5).

### The maximum von Mises stress and screw deformation for S1 and S2 segment screws

In all motion states, the maximum stress on all internal fixation devices remained below the yield strength of titanium alloy. In the standing position, the maximum von Mises

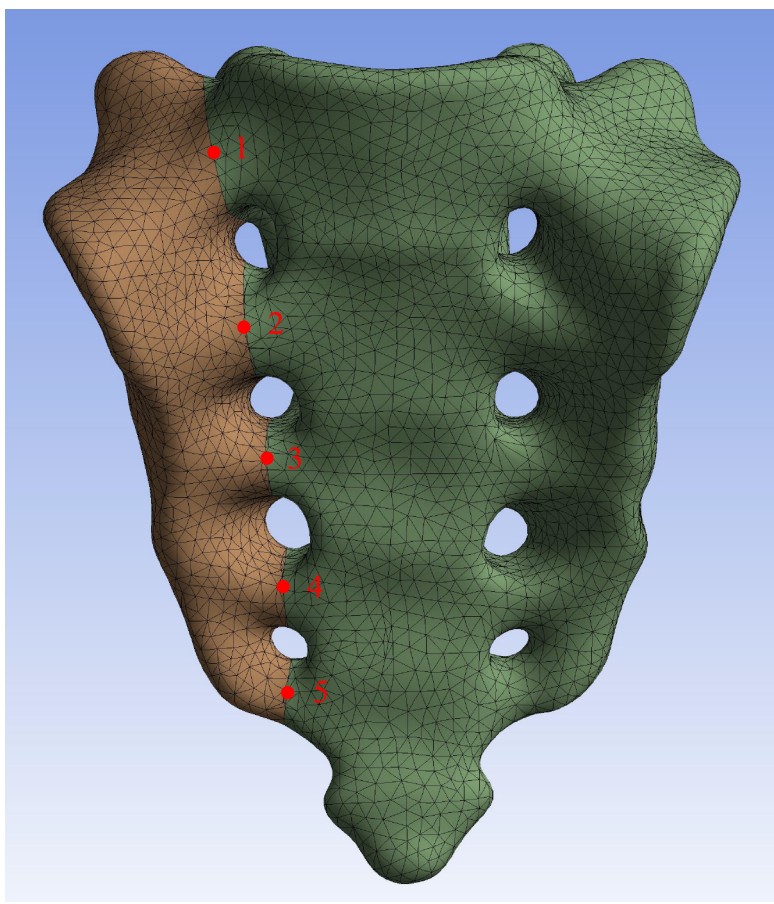

**Figure 4** Five pairs of relative displacement observation points were set on both sides of the fracture line on the anterior surface of the sacral.

stress in S1 segment screws varied significantly among groups: G1 had the highest stress (104.03 MPa), and G6 had the lowest (39.28 MPa). The stress values were ranked: G1 > G3 > G5 > G7 > G6. For S2 segment screws, the highest stress was in G2 (83.42 MPa) and the lowest in G7 (48.68 MPa), with the ranking: G2 > G4 > G5 > G3 > G6 > G7 (Fig. 6, Table 1).

The screw deformation analysis showed that for the S1 segment screws, G1 had the highest deformation (0.63 mm) and G5 had the lowest (0.30 mm), with the ranking: G1 > G3 > G6 > G7 > G5. For the S2 segment screws, G2 had the highest deformation (0.36 mm) and G5 had the lowest (0.14 mm), with the ranking: G3 > G2 > G6 > G7 > G5. In other motion states, the trends in maximum von Mises stress and deformation for both S1 and S2 segment screws were similar to those in the standing position (Table 1, Supplemental Information 1).

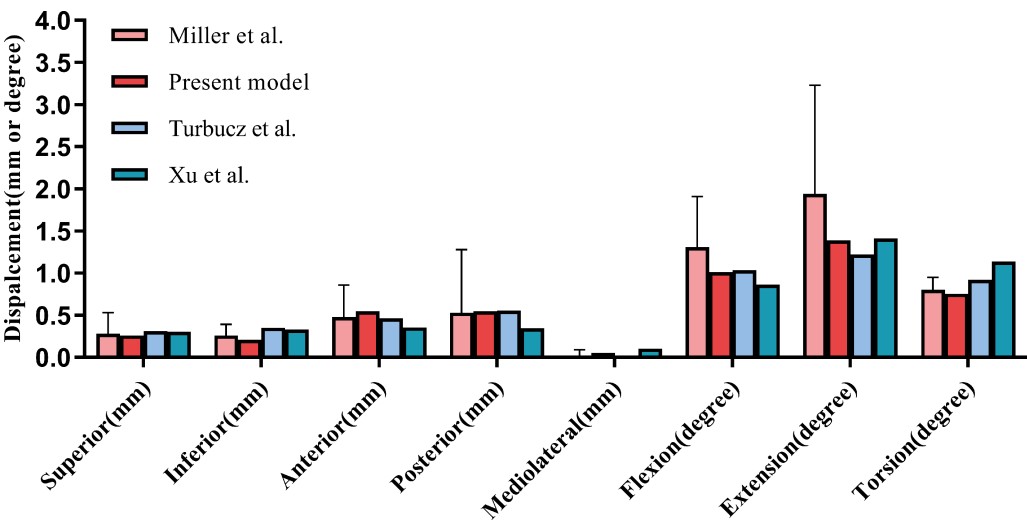

**Figure 5** Comparison of our finite element model with the experimental findings of *Miller, Schultz & Andersson (1987)* on sacral displacement under comparable loads. The error bar indicates one standard deviation.

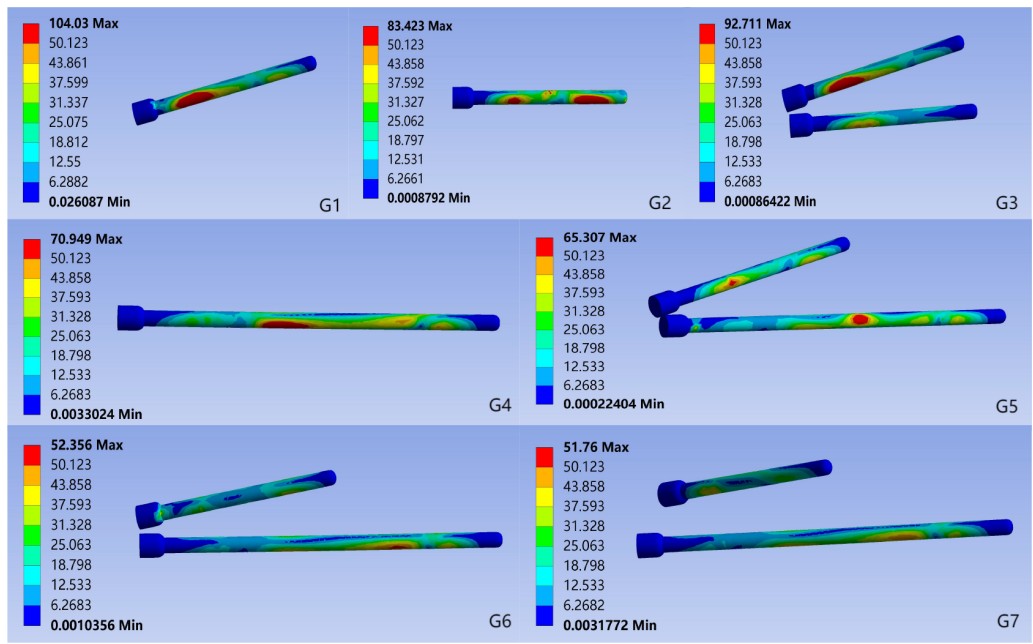

**Figure 6** Von Mises stress for each group of implants in the standing position. G1: S1-ISS; G2: S2-ISS; G3: S1-ISS+ S2-ISS; G4: S2-TTS; G5: Anterior S1-ISS + S2-TTS; G6: Middle S1- ISS + S2-TTS; G7: Posterior S1-ISS + S2-TTS.

## The vertical displacement and sagittal plane angular displacement of the superior surface of the S1

Under the six motion conditions, the trends in vertical displacement and sagittal angular displacement of the sacral upper surface were consistent among groups. G5 had the lowest

**Table 1   Under standing conditions, the maximum von Mises stress and deformation of the S1 and S2 segment screws.**

|  | Maximum von Mises stress of the screw (MPa) | | Deformation of the screw (mm) | |
|  | S1 segment | S2 segment | S1 segment | S2 segment |
| --- | --- | --- | --- | --- |
| G1 | 104.03 | – | 0.63 | – |
| G2 | – | 83.42 | – | 0.34 |
| G3 | 92.711 | 53.90 | 0.62 | 0.36 |
| G4 | – | 70.95 | – | 0.20 |
| G5 | 56.13 | 65.31 | 0.30 | 0.14 |
| G6 | 39.28 | 52.36 | 0.36 | 0.17 |
| G7 | 51.76 | 48.68 | 0.35 | 0.17 |

**Table 2   Under standing conditions, the vertical displacement and sagittal plane angular displacement of the superior surface of the S1.**

|  | Vertical displacement of the superior surface of S1 (mm) | | | The superior surface of S1 in the sagittal plane angular displacement (degree) |
|  | Min | Max | Mean | |
| --- | --- | --- | --- | --- |
| G1 | 0.4510 | 0.7004 | 0.5774 | 2.8784 |
| G2 | 0.4924 | 0.7421 | 0.6194 | 5.1653 |
| G3 | 0.4057 | 0.6794 | 0.5439 | 1.4063 |
| G4 | 0.3122 | 0.5082 | 0.4117 | 3.6868 |
| G5 | 0.2277 | 0.3594 | 0.2949 | 0.4404 |
| G6 | 0.2843 | 0.4368 | 0.3628 | 2.0354 |
| G7 | 0.2598 | 0.3990 | 0.3300 | 1.8314 |

values (vertical displacement: 0.2949 mm, sagittal angular displacement: 0.4404°), while G2 had the highest (vertical displacement: 0.6194 mm, sagittal angular displacement: 5.1653°). The vertical displacement of the upper surface of the sacral in each group, from highest to lowest, is as follows: G2 > G1 > G3 > G4 > G6 > G7 > G5. The sagittal plane angular displacement, from highest to lowest, is as follows: G2 > G4 > G1 > G3 > G6 > G7 > G5 (Table 2, Supplemental Information 2).

**Relative displacement of the observation point of the anterior sacral**
In the standing position, the relative displacement of the anterior sacral surface observation points varied significantly among groups. G5 had the smallest displacement (0.0700 mm), while G2 had the largest (0.1425 mm). The ranking from highest to lowest was: G2 > G4 > G1 > G3 > G7 > G6 > G5 (Fig. 7).

Under various motion conditions (flexion, left flexion, right flexion, left rotation), G2 had the highest relative displacement, while G5 had the lowest. During right rotation, G3 had the highest displacement, but G5 still had the lowest (Supplemental Information 3).

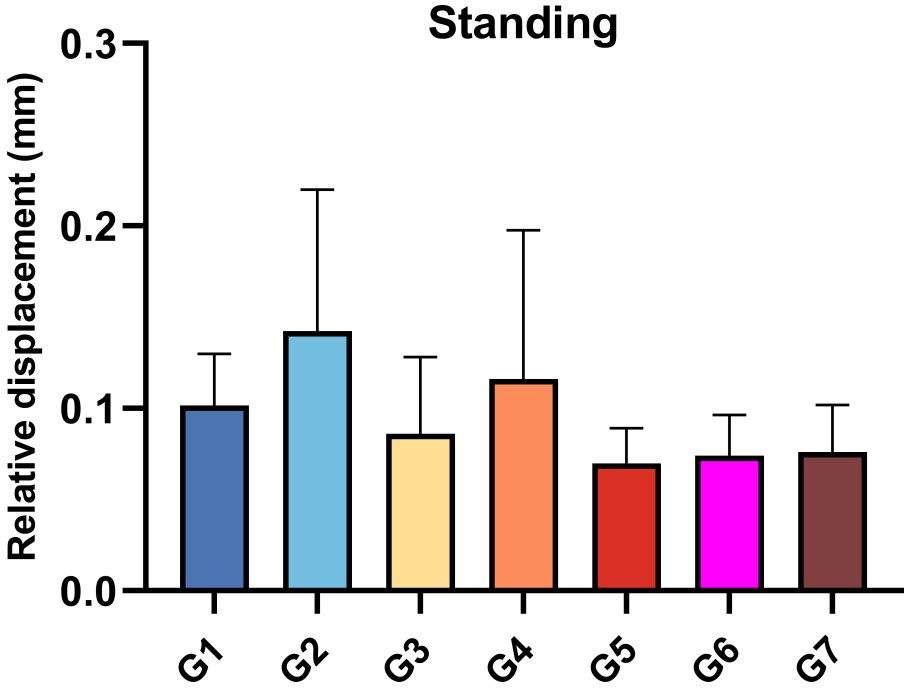

**Figure 7** **Relative displacement of the observation points on the anterior surface of the sacral when standing.** G1: S1-ISS; G2: S2-ISS; G3: S1-ISS+ S2-ISS; G4: S2-TTS; G5: Anterior S1-ISS + S2-TTS; G6: Middle S1- ISS + S2-TTS; G7: Posterior S1-ISS + S2-TTS.

## DISCUSSION

In this study, finite element analysis was conducted to investigate the biomechanical performance of different combinations of ISS and TSS for treating undisplaced Denis type II fractures in sacral dysmorphism. The results showed that G5 (anterior S1-ISS + S2-TSS) had superior biomechanical properties compared to other combinations, offering a new concept and reference for selecting optimal internal fixation strategies in clinical practice.

### Analysis of the biomechanical properties of internal fixation

The study showed that titanium's yield stress exceeds the maximum stress of screws in all internal fixation techniques, indicating a low risk of fatigue failure (*Fu et al., 2014*).

Additionally, the maximum von Mises stress in screws used with TSS is significantly lower than that in screws used with ISS alone. This suggests that TSS placement effectively reduces stress concentration and lowers the risk of fatigue failure, consistent with prior findings (*Turbucz et al., 2023*).

Screw deformation magnitude correlates with its dynamic behavior under cyclic loading. Excessive deformation can induce periodic micro-movement within the bone, leading to screw loosening or thread cutout (*Beucler, 2024*). In this study, greater screw deformation was observed when Denis II sacral fractures were fixed solely with ISS, potentially increasing the risk of loosening. However, combined ISS and TSS fixation significantly reduced screw

deformation, suggesting that TSS can enhance stability, minimize screw oscillation, and mitigate thread cutout.

## Sacral stability assessment

One of the main indications employed to evaluate the biomechanical stability of the sacral is the vertical displacement of the upper surface (*Fan et al., 2024*; *Turbucz et al., 2023*). Previous research has shown that a single TSS provides superior fixation compared to two ISS, significantly reducing sacral upper surface displacement, even at the S2 level (*Bradley et al., 2022*; *Zhao et al., 2013*). In this study, combined ISS and TSS fixation was more effective in reducing vertical displacement than TSS alone. ISS alone struggles to restrict sagittal rotation of the sacral upper surface under stress (*Beucler, 2025*; *Schildhauer et al., 2003*). TSS demonstrated superior anti-rotation capabilities compared to ISS alone, likely due to its immobilization of the sacrum and contralateral iliac cortex, providing more stable fixation. Combined ISS and TSS fixation further enhanced rotational stability by offering multiplanar support across two sacral segments.

## Stability of the fracture

The relative displacement of the observation point on the anterior sacral surface reflects the impact of internal fixation on fracture stability (*Zheng et al., 2021*). In this study, the combination of ISS and TSS exhibited the lowest and least deviated relative displacement among all groups. This indicates that ISS and TSS together provide optimal fracture stability.

In this study, the combination of ISS and TSS was found to provide excellent biomechanical and fracture stability in undisplaced sacral dysmorphic fractures. Specifically, the combination of anterior S1-ISS and S2-TSS demonstrated the highest stability. The potential reasons include: (1) Pelvic force conduction originates from the upper sacral endplate, travels through the sacral wing and sacroiliac joint to the arcuate line, and then descends. The anterior S1-ISS aligns more closely with the pelvic physiological structure. (2) The S1-ISS, located anteriorly, is closer to the medial iliac aspect and the anteromedial cortex of the sacral wing. According to Wolff's law, the elevated trabecular bone density along the screw trajectory results in stronger screw purchase.

## Limitation

Despite progress in this study, limitations remain. First, the placement of anterior S1-ISS is technically challenging. The complex sacral anatomy and proximity to major neurovascular structures make nail placement difficult with only X-ray fluoroscopy, often necessitating expensive technologies like computer navigation (Fig. 8). Second, the finite element analysis method used in this study provides theoretical mechanical predictions but lacks verification through cyclic loading tests on cadaver models. This limits the comprehensive evaluation of screw mechanical properties and stability. Future studies should use cadaver experiments to better replicate the actual biomechanical environment. Finally, the study's findings have not been clinically verified, and long-term radiological and functional assessments have not been conducted on patients. The feasibility, reproducibility, and long-term effects of the surgical construct under daily load must be evaluated in clinical practice. Follow-up

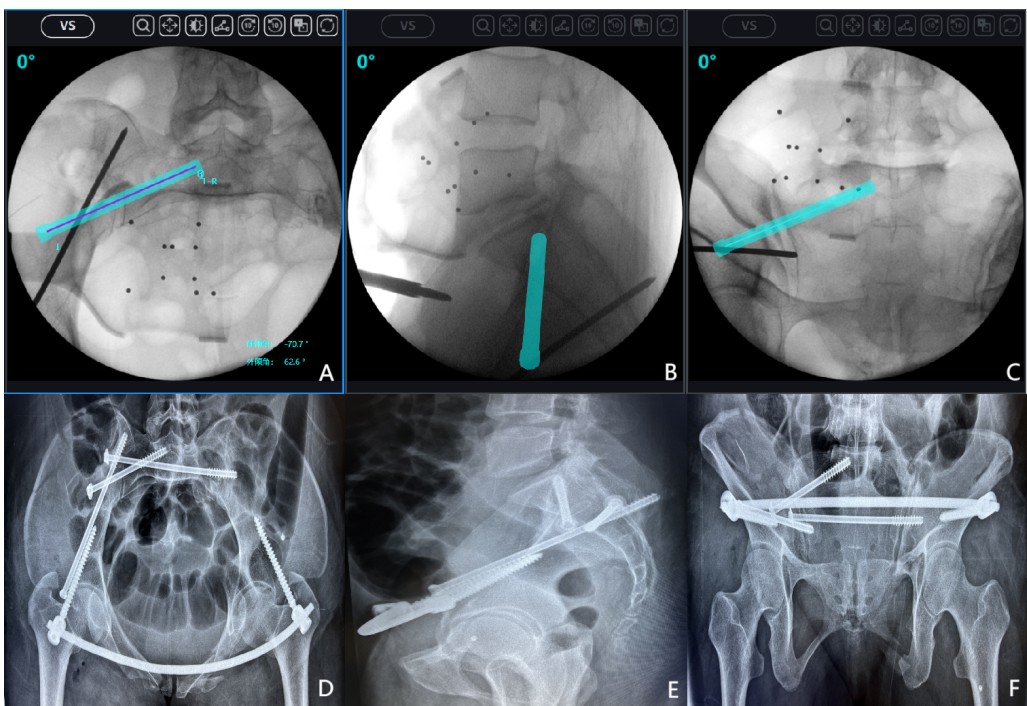

**Figure 8** **An example of anterior S1 iliosacral screws (S1-ISS) insertion under surgical robot navigation.** After successful closed reduction, the surgical robot was used to plan the appropriate screw trajectory on the inlet, outlet, and lateral sacral radiographs. (A) Intraoperative inlet pelvic radiograph; (B) Intraoperative lateral sacral radiograph; (C) Intraoperative outlet pelvic radiograph; (D) Postoperative inlet pelvic radiograph; (E) Postoperative lateral sacral radiograph; (F) Postoperative outlet pelvic radiograph.

research should include multicenter prospective clinical trials to determine the practical utility of the proposed surgical techniques.

## CONCLUSION

In this study, finite element analysis was conducted on seven internal fixation methods for treating Denis Type II sacral fractures. Based on our results, in patients with undisplaced vertical fracture of sacral dysmorphism, when it is not possible to safely insert a TTS into S1, the optimal combination is one ISS inserted into S1 and one TTS inserted into S2. Moreover, the combination of anterior S1-ISS and S2-TTS fixation demonstrated superior biomechanical stability compared to all other internal fixation methods.

**List of abbreviations**

| | |
|---|---|
| **ISS** | Iliosacral screw |
| **TTS** | Transiliac-transsacral screw |

### Funding

This work was supported by the National Natural Science Foundation of China (No. 81660366, 82160416), Guangxi Medical and health key discipline construction project and Guangxi Medical and health key cultivation discipline construction project. The funders had no role in study design, data collection and analysis, decision to publish, or preparation of the manuscript.

### Grant Disclosures

The following grant information was disclosed by the authors:
National Natural Science Foundation of China: 81660366, 82160416.
Guangxi Medical and health key discipline construction project.
Guangxi Medical and health key cultivation discipline construction project.

### Competing Interests

The authors declare there are no competing interests.

### Author Contributions

- Peishuai Zhao conceived and designed the experiments, analyzed the data, prepared figures and/or tables, authored or reviewed drafts of the article, and approved the final draft.
- Chengfei Peng conceived and designed the experiments, performed the experiments, authored or reviewed drafts of the article, and approved the final draft.
- Honghu Lin performed the experiments, prepared figures and/or tables, and approved the final draft.
- Ying Ji analyzed the data, authored or reviewed drafts of the article, and approved the final draft.
- Weiyi Pang analyzed the data, authored or reviewed drafts of the article, and approved the final draft.
- Chaoyong Bei analyzed the data, prepared figures and/or tables, authored or reviewed drafts of the article, and approved the final draft.

### Human Ethics

The following information was supplied relating to ethical approvals (i.e., approving body and any reference numbers):

The study was approved by the ethics committee of Guilin Medical University Affiliated Hospital (Number: 2024QTLL-02).

### Data Availability

The raw measurements are available in the Supplemental File.

The DOIs for the 44 individual MorphoSource entries are available in the Supplemental Files.

## Supplemental Information

Supplemental information for this article can be found online at http://dx.doi.org/10.7717/peerj.20139#supplemental-information.

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
