# Peer review of "Biomechanical analysis of iliosacral and transiliac–transsacral screw combinations for fixation of undisplaced Denis II vertical shear fractures in dysmorphic sacrum"

_PeerJ, doi:10.7717/peerj.20139_

## Round 0.1 · original submission · Major Revisions

Respond to all the comments from the reviewers. You do not need to add any of their suggested citations unless you feel they are necessary

Reviewer 1 ·

Basic reporting

See additional comments.

Experimental design

The design of the study is appropriate.

Validity of the findings

The results can be enhanced and supported by statistical correlation coefficient.

Additional comments

Title: do not use sign such as “/”.
The abstract is structured; add an introductory phrase before stating the objective; the keywords should be checked in accordance with MeSH.
The introduction should provide more examples experiments that assess screw position, in relation to the scientific literature (for e.g. https://doi.org/10.1016/j.promfg.2020.03.070 ). Also, the AO/OT acronym should be explained (including the abstract).
The methods are structured in subsections. Please provide the CT model used.
The results can be enhanced and supported by statistical correlation coefficient.
The discussions interpret the research results. Future research perspectives should be provided.
The conclusions are clear.
The references are adequate but can be extended as suggested above given the type of paper.

Reviewer 2 ·

Basic reporting

Thank you for reaching out to review this work.

The authors aimed to test a few number of different types of transiliosacral or transiliosacroiliac screw fixation techniques for percutaneous stabilization of Denis 2 vertical shear sacral fracture.

One of the main points of the authors' study is to provide this analysis for a dysmorphic sacrum with a peculiar slope of the sacral wing.

I have a few concerns:

(1) The type of dysmorphic sacrum that they present is not frequently encountered in clinical practice, even in specific population (10% in Japanese for example). If a spine surgeon meets such sacrum morphology, he / she will adapt the trajectory of his /her fixation to this precise patients, using already well known procedures such as transiliosacral or transiliosacroiliac percutaneous fixation, percutaneous triangular spinopelvic fixation, or open triangular bilateral iliolumbar fixation.
Surgeons may use radiologic guidance available and/or best suited for them, between double oblique fluoroscopy-guides view or intraoperative CT navigation, to complete the procedure safely.
Relevant references:
-Iga T. Iliosacral screw corridors in Japanese subjects: a study using reconstruction CT scans. OTA Int. 2021 Aug 6;4(3):e145. doi: 10.1097/OI9.0000000000000145. PMID: 34746676; PMCID: PMC8568404.

(2) Transiliosacral or transiliosacroiliac screwing is well adapted for horizontal stability, but less suited for vertical shearing forces and even more for rotation nutation forces which happen when patients walk.
Relevant references:
-Ziran N, Collinge CA, Smith W, Matta JM. Trans-sacral screw fixation of posterior pelvic ring injuries: review and expert opinion. Patient Saf Surg. 2022 Jul 27;16(1):24. doi: 10.1186/s13037-022-00333-w. PMID: 35897108; PMCID: PMC9327417.
-Schildhauer TA, Ledoux WR, Chapman JR, Henley MB, Tencer AF, Routt ML Jr. Triangular osteosynthesis and iliosacral screw fixation for unstable sacral fractures: a cadaveric and biomechanical evaluation under cyclic loads. J Orthop Trauma. 2003 Jan;17(1):22-31. doi: 10.1097/00005131-200301000-00004. PMID: 12499964.

(3) The authors did not run any loading test under rotation / nutation condition, which possibly denotes a lack of understanding of pelvic biomechanics because there forces apply when patients walk.
Relevant references:
-Beucler N. Triangular spinopelvic fixation for U-shaped sacral fractures and tile C pelvic disruptions: counter-nutation (anteflexion and rotation) load-bearing instability requires complementary anterior pelvic ring fixation. Neurosurg Rev. 2024 Aug 1;47(1):389. doi: 10.1007/s10143-024-02650-3. PMID: 39085443.

(4) I wonder how the G5 model S1 transiliosacral screw could be inserted in real life. You would have to get through 20cm of gluteal muscle, and I am not sure whether the sacral part of the trajectory would not cross the path of S1 nerve root, which displays an oblique anterior, exterior, and inferior direction

(5) The type of surgical fixation must be adapted to the type of pelvic trauma:
-transiliosacral / transiliosacroiliac fixation for undisplaced sacro-iliac dissociation, or sacral Denis type 1 or 2 vertical fracture,
-triangular fixation including iliolumbar fixation and transsacroiliac screw for horizontally and vertically unstable sacro-iliac dissociation, or sacral Denis type 1 or 2 vertical fracture,
-open triangular fixation for more complex trauma, or Denis type 3 injuries.

(6) It has already been shown that transiliosacroiliac screws provide more 3D stability compared to only transiliosacral screws.
Relevant references:
-Cintean R, Fritzsche C, Zderic I, Gueorguiev-Rüegg B, Gebhard F, Schütze K. Sacroiliac versus transiliac-transsacral screw osteosynthesis in osteoporotic pelvic fractures: a biomechanical comparison. Eur J Trauma Emerg Surg. 2023 Dec;49(6):2553-2560. doi: 10.1007/s00068-023-02341-6. Epub 2023 Aug 3. PMID: 37535095; PMCID: PMC10728224.

Conclusion:
The authors must be commended for conducting this finite element analysis.
Nevertheless, a few elements discussed above could allow for a better translation into clinical practice.

I would be pleased to review a revised version of this work if the authors are willing to put in the work required.

All the best.

Experimental design

see above

Validity of the findings

see above

Additional comments

see above

---

## Round 0.2 · Major Revisions

Reviewer 2 still has some substantial comments.

Reviewer 1 ·

Basic reporting

The reporting is clear.

Experimental design

The design of the study is adequate.

Validity of the findings

The results are novel.

Additional comments

The authors have improved their paper accordingly.

Reviewer 2 ·

Basic reporting

Thank you for reaching out again to oversee this revision.

The authors have partially addressed my comments.

(1) Title should include «  of undisplaced Denis 2 vertical shear fracture in dysmorphic sacrum »

(2) They should include the radiographs that they have provided in answer C4 which displays their clinical experience and greatly illustrates the G5 model oblique S1 transiliosacral screw.

(3) I still disagree with the fact that sole combination of iliosacral and trans-ilio-sacro-iliac screws provide rotational stability
-Ziran N, Matta JM et al., Transsacral screw fixation of posterior pelvic ring injuries: review and expert opinion. Patient Saf Surg 1, 2022. Under creative common license 4.0
-Schildhauer TA, Ledoux WR, Chapman JR, Henley MB, Tencer AF, Routt ML Jr. Triangular osteosynthesis and iliosacral screw fixation for unstable sacral fractures: a cadaveric and biomechanical evaluation under cyclic loads. J Orthop Trauma. 2003 Jan;17(1):22-31. doi: 10.1097/00005131-200301000-00004. PMID: 12499964.

(4) Discussion should be divided into distinct paragraphs with specific headings in order to ease reading

(5) Discussion should be shortened / summarized. It is too lengthy in its current form.

(6) discussion part « vertical displacement of the upper surface of the sacrum is often used as a key indicator »
Pelvic incidence is also a major indicator of proper reduction of sacral trauma
-Introducing the relationship between pelvic incidence and subtype of U-shaped sacral fracture according to Roy-Camille classification: How to restore sagittal balance in spinopelvic trauma. Injury. 2025 Mar 24:112286. doi: 10.1016/j.injury.2025.112286. Epub ahead of print. PMID: 40157870.

(6) Limitations should include:
-the possible technical difficulty of placing G5 S1 iliosacral screws,
-the absence of laboratory testing on cadaveric anatomical testing,
-the need to test the feasibility and reproducibility of these theoretical surgical constructs in the real clinical settings, and the long-term radiologic and functional outcome of real life patients who sustain daily life load bearing situations

All the best .

Experimental design

-

Validity of the findings

-

Additional comments

-

---

## Round 0.3 · Major Revisions

Reviewer 2 ·

Basic reporting

Thank you for reaching out to review this work.

The authors have addressed some of my comments, but lots of comments have not been properly addressed.

Comment 1:
Pioneer references lack in this manuscript:

(1) Pathophysiology of sacrum trauma:
-Roy-Camille R, Saillant G, Gagna G, Mazel C. Transverse fracture of the upper sacrum. Suicidal jumper's fracture. Spine (Phila Pa 1976). 1985 Nov;10(9):838-45. doi: 10.1097/00007632-198511000-00011. PMID: 4089659.
-Denis F, Davis S, Comfort T. Sacral fractures: an important problem. Retrospective analysis of 236 cases. Clin Orthop Relat Res. 1988 Feb;227:67-81. PMID: 3338224.
-Tile M. Pelvic ring fractures: should they be fixed? J Bone Joint Surg Br. 1988 Jan;70(1):1-12. doi: 10.1302/0301-620X.70B1.3276697. PMID: 3276697.

(2) Surgical technique:
-Schildhauer TA, Ledoux WR, Chapman JR, Henley MB, Tencer AF, Routt ML Jr. Triangular osteosynthesis and iliosacral screw fixation for unstable sacral fractures: a cadaveric and biomechanical evaluation under cyclic loads. J Orthop Trauma. 2003 Jan;17(1):22-31. doi: 10.1097/00005131-200301000-00004. PMID: 12499964.
-Nork SE, Jones CB, Harding SP, Mirza SK, Routt ML Jr. Percutaneous stabilization of U-shaped sacral fractures using iliosacral screws: technique and early results. J Orthop Trauma. 2001 May;15(4):238-46. doi: 10.1097/00005131-200105000-00002. PMID: 11371788.

(3) Reduction criteria:
-Matta JM, Tornetta P 3rd. Internal fixation of unstable pelvic ring injuries. Clin Orthop Relat Res. 1996 Aug;(329):129-40. doi: 10.1097/00003086-199608000-00016. PMID: 8769444.
-Beucler N. Introducing the relationship between pelvic incidence and subtype of U-shaped sacral fracture according to Roy-Camille classification: How to restore sagittal balance in spinopelvic trauma. Injury. 2025 Mar 24:112286. doi: 10.1016/j.injury.2025.112286. Epub ahead of print. PMID: 40157870.

Comment 2:
manuscript is still much to lenghty

Comment 3:
sacrum dismorphism rate is exagerated in this manuscript. The authors dont need to exagerate the rate of sacral dismorphism to justify their study.

Comment 4:
The authors need to highlight that in rather stable undisplaced Denis 2 sacral fracture, which I personally manage with orthopedic treatment with 6 weeks of bed rest and then 6 weeks of sitting position, the major ligaments of the pelvis still hold and thus participate in the inherent stability of the fracture. The percutaneous screws are only a temporary supplementary support here.
Did the authors take the partial resistance of iliosacral, sacrotuberous, and sacrospinous intact ligaments into account during stress tests?

Conclusion:
This manuscript still requires some work to become clinically relevant.

All the best

Experimental design

-

Validity of the findings

-

Additional comments

-

---

## Round 0.4 · accepted · Accept

Dear Authors,

I am writing with pleasure that your manuscript is accepted for publication. Since, this is only an editorial acceptance and need few tasks to be completed, I will advise you to be available for few days to avoid any delays.

All the best

·

Basic reporting

The article uses clear and professional English throughout. Technical terminology is employed appropriately for the target audience, and the tone remains formal and academic across all sections. Sentences are concise and well-structured, enhancing the readability of complex concepts. Key prior studies are cited appropriately, and the literature review effectively identifies the knowledge gap that the current research aims to address. References are relevant, up-to-date, and sufficiently comprehensive to position the study within the broader research landscape. The results are directly relevant and are statistically and conceptually sound. Data interpretation is justified, and the conclusions are supported by the evidence presented. Potential limitations are acknowledged, and the scope of the findings is clearly defined.

Experimental design

The article represents original primary research that clearly aligns with the Aims and Scope of the journal. The introduction explicitly states how the study addresses a recognized knowledge gap, thereby justifying the significance and necessity of the work.

Experimental procedures are appropriate for the research question and are executed with precision and integrity. The ethical considerations related to the study are acknowledged and appropriately handled, including any necessary approvals or informed consent procedures. The methodological section is thorough and provides sufficient detail to enable reproducibility by other researchers in the field.

Validity of the findings

The article presents a well-executed primary research study focused on addressing a clearly articulated and meaningful research question.

The data are robust, statistically sound, and appropriately controlled, ensuring confidence in the reliability of the findings.

The conclusions are clearly stated and appropriately limited to the scope of the supporting results. The authors refrain from overgeneralization, maintaining a high standard of scientific integrity.